# Gait Speed as a Biomarker of Cognitive Vulnerability: A Population-Based Study with Cognitively Normal Older Adults

Marcelo de Maio Nascimento [1,*] , Élvio Rúbio Gouveia [2,3,4], Adilson Marques [5,6] , Bruna R. Gouveia [3,4,7,8], Priscila Marconcin [9,10] and Andreas Ihle [4,11,12]

1 Department of Physical Education, Federal University of Vale do São Francisco, Petrolina 56304-917, Brazil
2 Department of Physical Education and Sport, University of Madeira, 9020-105 Funchal, Portugal; erubiog@staff.uma.pt
3 LARSyS—Laboratory for Robotics and Engineering Systems, Interactive Technologies Institute, 9020-105 Funchal, Portugal; bgouveia@esesjcluny.pt
4 Center for the Interdisciplinary Study of Gerontology and Vulnerability, University of Geneva, 1205 Geneva, Switzerland; andreas.ihle@unige.ch
5 CIPER—Interdisciplinary Centre for the Study of Human Performance, Faculty of Human Kinetics, University of Lisbon, 1495-751 Lisbon, Portugal; amarques@fmh.ulisboa.pt
6 ISAMB—Environmental Health Institute, Faculty of Medicine, University of Lisbon, 1649-020 Lisbon, Portugal
7 Regional Directorate of Health, Government of the Autonomous Region of Madeira, 9004-515 Funchal, Portugal
8 Saint Joseph of Cluny Higher School of Nursing, 9050-535 Funchal, Portugal
9 Faculty of Human Kinetics, University of Lisbon, 1495-751 Lisbon, Portugal; priscilamarconcin@fmh.ulisboa.pt
10 KinesioLab, Research Unit in Human Movement Analysis, Piaget Institute, 2805-059 Almada, Portugal
11 Department of Psychology, University of Geneva, 1205 Geneva, Switzerland
12 Swiss National Centre of Competence in Research LIVES-Overcoming Vulnerability: Life Course Perspectives, 1015 Lausanne, Switzerland
* Correspondence: marcelo.nascimento@univasf.edu.br; Tel.: +55-(87)-2101-6856

**Abstract:** We aimed to examine associations between cognitive vulnerability and gait speed (GS) in a large older sample. A cross-sectional study analyzed data from the "Health, Lifestyle and Fitness in Adults and Seniors in Amazonas" (SEVAAI) project. In total, 697 participants were included (mean age 70.35 ± 6.86 years). Usual and fast GS were evaluated, and cognitive performance was examined by the COGTEL test battery. There was a positive and large correlation between cognition (COGTEL score) and usual GS ($r = 0.510$; $p < 0.001$) and fast GS ($r = 0.503$; $p < 0.001$). The usual GS, as a continuous variable, indicated a chance of improved cognitive performance by up to 55%, and fast GS by up to 82%. After controlling for potential confounders (i.e., sex, age, MMSE and years of education), usual and fast GS indicated a chance of improving cognition, respectively, in 57% and 85%. Analysis of GS in quartiles (Q) showed high and significant associations between usual and fast GS and cognitive vulnerability. GS classified as Q1 (slower), Q2 and Q3 represented a greater chance of presenting cognitive deficits, respectively, than in participants with both GS classified as Q4 (highest). Cognitive vulnerability was associated with low GS. Usual and fast GS can be used as complementary measures for the evaluation of cognitively normal Brazilian older adults.

**Keywords:** gait speed; cognition; older adult; biomarker

## 1. Introduction

Gait and cognition are interrelated systems [1,2]. The simple act of walking requires the complex integration of external sensory information, which requires precise commands from the central nervous system. As a high motor action, gait is subordinated to neural networks linked to the cortical and subcortical structures of the spinal cord [3,4]. Thus, due to the need for brain negotiation involved in gait [5], gait speed (GS) alterations can be

used as a non-invasive early indicator to detect pathogenic processes associated with brain functioning in the older adult population [6–8]. Therefore, individuals with slower GS are considered more likely to experience a cognitive decline (e.g., dementia) than those with faster GS [9–12].

From the relationship between the quality of gait (motor action) and the substrates involved in cognitive processes, it is possible to build a complementary approach to understand the functioning and/or alteration of neural functions during old age [13–15]. GS deficits may reflect the age-related atrophy of shared neural structures and/or greater dependence on cognitive processes compromised by sensorimotor integration [13]. Studies carried out in older adults using magnetic resonance imaging, nuclear imaging and electroencephalography revealed associations between poor gait performance and structural alterations in several subregions of the frontal lobe [16,17]. Moreover, reductions in grey matter volume were observed—more specifically, in the prefrontal cortex and hippocampus [18] and in the subcortical regions of the basal ganglia and cerebellum [19]. Quantitative gait changes have also been associated with white matter hyperintensities, increased ventricular volume and atrophy of the primary motor cortex [20].

In general, the gait exam is a simple and quick screening test, which can be performed in an outpatient setting without the need for expensive materials. GS is a sensitive measure that allows multiple uses, such as assessing the probability of older adult falls [21], as well as the estimation of the frailty of older people when associated with other indicators [22]. In recent decades, a considerable number of population studies have used the GS (in quantitative analyses) as a biomarker (screening) of cognitive vulnerability in older adults [3,7,23]. The GS examination is considered a substantial measure due to the difficulty in detecting superficial cognitive impairments [24]. In the clinical area, screening for mild cognitive impairment (MCI) or the beginning of the process is a major challenge [13,25,26]. Notably, few studies have examined the association between GS and cognitive vulnerability in cognitively normal older adults, especially in a population-based study [6,27,28]. The majority of cross-sectional investigations have included older adults with cognitive impairments (e.g., Parkinson's disease, Alzheimer's, dementia). Moreover, due to methodological differences between existing cross-sectional studies, there is a low consensus on the associations between cognitive impairment and GS, especially in the single-task condition [26]. When dealing with the older and cognitively normal population, cross-sectional studies have the advantage of helping to identify cognitive alterations during the genesis of the process (pre-clinical phases), a propitious moment for an early therapeutic intervention. In addition, there is a lack of information on the relationship between GS and cognitive change in normal ageing, especially in older Latin American populations [8].

Although, in recent decades, studies have examined the association between GS and cognitive impairment with cognitively normal older adults from different countries, such as the United States [25,26,29], Canada [19,20] and Japan [30], to date, we are not aware of population-based investigations that have investigated associations between GS and cognitive deficits in cognitively normal and independent Brazilian older adults. Information based on the geolocation of the cohort is important, as it provides a deeper understanding of the facets of the ageing process, contributing to the creation or restructuring of specific health policies/strategies. According to the Brazilian Society of Geriatrics and Gerontology [31], compared to other regions of the country, the older population of the northern region is ageing in conditions of greater vulnerability: the process takes place in a scenario of socio-economic fragility, with worse access to basic care and education.

Thus, we used data from the Health, Lifestyle and Physical Fitness in Adults and Elderly in Amazonas (SEVAAI) research project to examine associations between cognitive vulnerability and GS in a large older sample. We hypothesized that (1) higher GS values are positively associated with better cognitive performance, and (2) low GS is associated with cognitive vulnerability in cognitively normal older adults.

## 2. Materials and Methods

### 2.1. Study Design and Participants

This was a cross-sectional study carried out with 701 participants (433 women and 268 men). Four participants were excluded due to neurological diseases such as Alzheimer's (*n* = 3) and Parkinson's disease (*n* = 1). Participants were part of the research project "Health, Lifestyle and Fitness in Adults and Seniors in Amazonas" (SEVAAI). All were volunteers with unique geographic and cultural characteristics of the city of Manaus (municipalities of Fonte Boa and Apuí), located in the state of Amazonas, in the northern region of Brazil. Data were collected between 2016 and 2017.

The inclusion criteria were: (1) living in one of the three cities in the city of Manaus mentioned above; (2) minimum age of 60 years; (3) being able to walk independently and perform physical assessments; (4) present autonomy and independence in carrying out activities of daily living. The exclusion criterion were: (1) a score of <15/30 points in the Mini-Mental State Examination (MMSE)—the criterion was assumed as the limit of the inability to understand and follow the evaluation protocol of this study [32]—and (2) medical contraindication for the practice of exercise. All participants gave informed consent, and the present study included adherence to the declaration of Helsinki, which had been approved by the local ethics commission (CAAE: 56519616.0000.5016; N° 1.599,258; Brazil Platform).

### 2.2. Data Collection

#### 2.2.1. Cognitive Assessment

The participants' cognitive performance was individually assessed in a face-to-face interview, carried out by members of the field team trained in the application of protocols. The instrument used was the COGTEL, composed of six subtests [33]. The COGTEL includes important domains of cognitive function: (1) prospective memory; (2) short-term verbal memory; (3) working memory; (4) inductive reasoning; (5) verbal fluency and (6) long-term verbal memory. The calculation of the total score (continuous scale) was performed by the following formula: COGTEL total score = $7.2 \times$ prospective memory + $1.0 \times$ short-term verbal memory + $0.9 \times$ long-term verbal memory + $0.8 \times$ working memory + $0.2 \times$ verbal fluency + $1.7 \times$ inductive reasoning. In the Brazilian population, the psychometric properties of COGTEL were previously verified in an older adult sample [34], showing excellent test–retest reliability (0.85; *p* < 0.001), with high MMSE convergent validity (0.93; *p* < 0.001).

#### 2.2.2. Gait Speed

GS was assessed using the 30-Foot Walk Test [35]. First, participants were asked to walk a distance of 30 feet at their usual speed. After, they were asked to repeat the same test at a fast speed but without running in a second moment. Three attempts were measured for each gait type (usual speed and fast speed), and the best performance was assumed.

#### 2.2.3. Covariates

In a face-to-face interview, the participants reported sex, age, years of education, comorbidities (coronary heart, cerebrovascular, muscle diseases, depression and others), the number of types of medication consumed daily and total falls in the last 12 months. Body mass and height were measured using an anthropometric scale and a Welmy® stadiometer coupled with 0.1 cm and 0.1 kg [36]. Body mass index (BMI) was defined as (weight [kg])/ (height [$m^2$]).

#### 2.2.4. Statistical Analysis

To assess the normal distribution of the variables, the Kolmogorov–Smirnov test was used. Thus, metric variables were reported by medians (IQR) and categorical by frequencies (%). Differences between groups were processed by nonparametric Mann–Whitney U test or chi-square test for continuous or categorical variables, respectively. The cutoff point

used for composition (segregation) of the groups was determined by the overall mean of the total COGTEL (score = 18.9 points) minus 1 standard deviation. The literature indicated this procedure to identify individuals with cognitive vulnerability and, e.g., mild cognitive impairment [37]. The groups were established as follows: cognitive impairment with ≤17.9 points vs. normal cognition with ≥17.9 points. To test cross-sectional associations, we performed a three-phase approach. First, we tested whether higher GS values were positively associated with higher cognitive performance (first hypothesis). Thus, we examined the correlation of cognition (COGTEL total score) with usual GS and fast GS (motor performance). Correlations were presented in a scatter plot and interpreted by Pearson's correlation coefficients ($r$): 0.1 = small, 0.3 = medium and ≥0.5 = large [38].

Second, we verified whether GS as a continuous variable can predict cognitive vulnerability in cognitively normal older adults (second hypothesis). Thus, we chose for analysis the group "cognitive impairment" as a reference and developed two linear logistic regressions separately for gait: (1) we analyzed the performance of COGTEL (dependent variable, binary) vs. usual GS (independent variable), and (2) we examined the performance of COGTEL (dependent variable, binary) vs. fast GS (independent variable). Third, we verified whether slow GS increases the chance of cognitive vulnerability in cognitively normal older adults. In this case, we chose to analyze the "normal cognition" group as a reference, and then two multinomial logistic regressions for gait were performed separately: (1) we included the performance of COGTEL (dependent variable, binary) vs. usual GS (independent variable/quartile analysis), and (2) we chose the performance of COGTEL (dependent variable, binary) vs. fast GS (independent variable/quartile analysis). The use of quartiles (Q) was assumed to avoid the assumption of collinearity in both GS assessments. The highest quartile (fourth) was the reference category. Only covariates with a $p$-value < 0.20 in the univariate analysis were included as control factors in the adjusted analyses. Both the results of the logistic regression analysis and the multinomial regression analysis were presented in two models: Model 1 unadjusted and Model 2 adjusted for sex, age, MMSE and years of education. In all analyses, the odds ratios (ORs) and their respective confidence intervals (CI) were estimated. The statistical analyses were performed using IBM-SPSS (IBM Corp., Armonk, NY, USA), version 22.0. The significance level was defined as $\alpha = 0.05$.

### 3. Results

*3.1. Main Characteristics of the Participants*

Six hundred and ninety-seven participants were evaluated (see Table 1 for an overview). Of these, 331 were classified with cognitive impairment, and the other 366 were classified with normal cognitive performance. The median age of the entire sample was 69.2 (60.00–91.84) years; members of the cognitive impairment group indicated a median of 79.8 (60.24–91.84) years, while participants of the normal cognition group indicated 67.9 (60.00–89.46) years ($p < 0.001$). Regarding gender, 61.7% of the participants were women ($p = 0.002$). Significant differences were also found for BMI, years of education and the MMSE ($p < 0.001$). Among the self-reported comorbidities, the most prevalent was visual impairment (83.5%), followed by heart disease (56.7%). However, only musculoskeletal disease (42.3%) showed a statistical difference ($p < 0.001$).

**Table 1.** Main characteristics for the entire study population, according to cognitive performance on the COGTEL test.

| Variable | Full Sample (*n* = 697) | Cognitive Impairment (*n* = 331) | Normal Cognition (*n* = 366) | *p*-Value |
|---|---|---|---|---|
| Age in years (median, IQR) | 69.2 (60.00–91.84) | 79.8 (60.24–91.84) | 67.9 (60.00–89.46) | |
| 60–69 *n* (%) | 339 (48.6) | 139 (42.0) | 200 (54.6) | <0.001 |
| 70–79 *n* (%) | 274 (39.3) | 136 (41.1) | 138 (37.7) | |
| ≥80 *n* (%) | 84 (12.1) | 56 (16.9) | 28 (7.7) | |
| Gender *n* (%) women | 430 (61.7) | 187 (55.7) | 243 (67.3) | 0.002 |
| BMI (kg/m$^2$) (median, IQR) | 27.6 (16.35–47.48) | 26.9 (16.36–47.00) | 28.5 (16.35–47.48) | <0.001 |
| Years of education (median, IQR) | 4.0 (0.00–23.00) | 0.0 (0.00–16.00) | 8.0 (0.00–23.00) | <0.001 |
| Number of meds (median, IQR) | 2.0 (0.00–13.00) | 2.0 (0.00–10.00) | 1.5 (0.00–13.00) | 0.190 |
| Falls *n* (%) | 227 (32.38) | 111 (33.5) | 116 (31.7) | 0.605 |
| MMSE (median, IQR) | 25.0 (11.00–30.00) | 22.0 (11.00–30.00) | 27.0 (15.00–30.00) | <0.001 |
| Comorbidities *n* (%) | | | | |
| Heart disease | 395 (56.7) | 191 (57.7) | 204 (44.3) | 0.328 |
| Hearing impairment | 180 (25.8) | 81 (24.5) | 99 (27.0) | 0.245 |
| Visual impairment | 582 (83.5) | 268 (81.0) | 314 (85.8) | 0.054 |
| Musculoskeletal disease | 295 (42.3) | 112 (33.8) | 183 (50.0) | <0.001 |
| Cerebrovascular disease | 32 (4.6) | 16 (4.8) | 16 (4.4) | 0.455 |
| Depression | 32 (4.6) | 17 (5.1) | 15 (4.1) | 0.318 |

IQR = interquartile range; m = meter; s = second; MMSE = Mini-Mental State Examination; BMI = body mass index; $p < 0.05$.

### 3.2. Performance of Usual GS and Fast GS

Table 2 presents the gait performance results. The usual and fast GS analysis as continuous variables showed greater speed for the group with normal cognition ($p < 0.001$). The analysis by quartiles revealed a prevalence of low velocity for group members with cognitive impairment. Significant results were observed for usual GS in Q3 ($p = 0.003$), and for fast GS in Q1 ($p = 0.014$) and Q4 ($p = 0.028$).

**Table 2.** Performance of usual GS and fast GS (continuous variable), and performance prevalence in quartiles, according to the cognitive assessment of the COGTEL test.

| Variable | Full Sample (*n* = 697) | Cognitive Impairment (*n* = 331) | Normal Cognition (*n* = 366) | *p*-Value |
|---|---|---|---|---|
| Gait usual speed (m/s) (median, IQR) | 1.23 (0.35–3.20) | 1.08 (0.35–2.62) | 1.47 (0.55–3.20) | <0.001 |
| Gait fast speed (median, IQR) | 1.64 (0.45–4.17) | 1.45 (0.45–3.44) | 1.99 (0.73–4.17) | <0.001 |
| Gait usual speed *n* (%) | | | | |
| 1 (lowest) | 157 (22.5) | 121 (36.0) | 36 (0.9) | 0.195 |
| 2 | 191 (27.4) | 116 (32.1) | 75 (20.7) | 0.270 |
| 3 | 173 (24.8) | 67 (19.9) | 106 (29.3) | 0.003 |
| 4 (highest) | 176 (25.2) | 27 (8.0) | 149 (41.2) | 0.299 |
| Gait fast speed *n* (%) | | | | |
| 1 (lowest) | 162 (23.2) | 115 (34.7) | 47 (12.8) | 0.014 |
| 2 | 180 (25.8) | 112 (33.8) | 68 (18.6) | 0.117 |
| 3 | 176 (25.3) | 76 (23.0) | 100 (27.3) | 0.470 |
| 4 (highest) | 179 (25.7) | 28 (8.5) | 151 (41.3) | 0.028 |

IQR = interquartile range; m = meter; s = second; $p < 0.05$.

Figure 1 shows the performance of the usual and fast GS of both groups in the quartiles. Significant differences were found for usual GS (Figure 1A) in Q3 ($p = 0.003$), and for fast GS (Figure 1B) in Q1 ($p = 0.014$) and Q4 ($p = 0.028$). The analysis also showed differences in gait performance between groups for each quartile; the results are represented as Δ-Delta values.

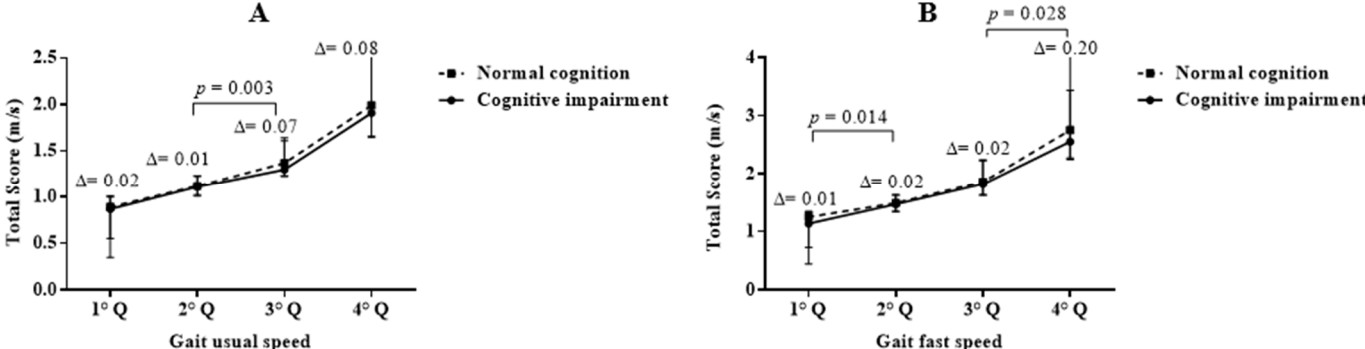

**Figure 1.** Comparative graph of the usual GS and GS fast (mean and standard errors), between the groups with and without cognitive impairment, according to velocity quartiles. Each consecutive top quartile of speed represents a better gait. (**A**) usual GS; (**B**) fast GS; m = meter; s = second; Q = quartile; $p < 0.05$.

### 3.3. Correlations between Usual and Fast GS and Cognitive Performance

Figure 2 shows the scatter plots of the correlation analysis between cognition (COGTEL total score) and the two GS performance assessments (as a continuous variable). Positive and large correlations were found between COGTEL and usual GS ($r = 0.510$; $p < 0.001$) (Figure 2A), as well as COGTEL and fast GS ($r = 0.503$; $p < 0.001$) (Figure 2B).

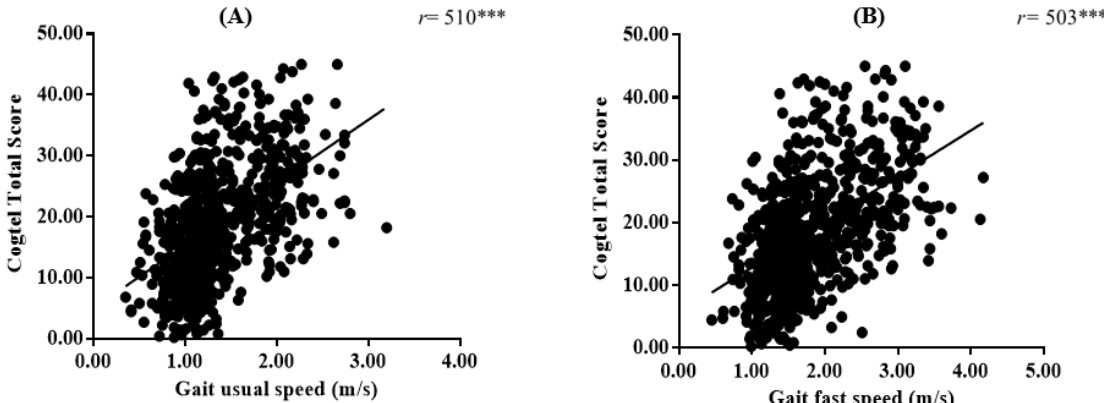

**Figure 2.** Scatter plot of correlations between usual GS and fast GS and cognitive performance in the COGTEL test. (**A**) usual GS; (**B**) fast GS; m = meter; s = second; *** $p < 0.001$.

### 3.4. Associations between Usual GS and Fast GS (Continuous Variable) and Cognitive Vulnerability

According to Table 3, the unadjusted analysis indicated a statistically significant negative association between cognition and usual GS (OR = −0.45, CI −0.553–0.410, $p < 0.001$) and fast GS (OR = −0.433, CI −0.384–0.280, $p < 0.001$). After controlling for potential confounders (i.e., sex, age, MMSE and years of education), the associations between cognition and GS were negative and attenuated, remaining significant: usual GS (OR = −0.180, CI −0.264–0.120, $p < 0.001$) and fast GS (OR = −0.150, CI −0.166–0.063, $p < 0.001$).

**Table 3.** Associations between gait speed (continuous and quartiles) and cognitive performance (COGTEL); results expressed by logistic and multivariate regression.

| Variable | Unadjusted OR (95% CI) | *p*-Value | Adjusted OR (95% CI) | *p*-Value |
|---|---|---|---|---|
| Usual gait [c] (m/s) * | −0.451 (−0.553–0.410) | <0.001 | −0.180 (−0.264–0.120) | <0.001 |
| Fast speed [c] (m/s) * | −0.433 (−0.384–0.280) | <0.001 | −0.150 (−0.166–0.063) | <0.001 |
| Usual speed (m/s) † | | | | |
| Quartile 4 (highest) | 1 | | 1 | <0.001 |
| Quartile 3 | 3.488 (2.091–5.817) | <0.001 | 1.773 (0.953–3.297) | 0.071 |
| Quartile 2 | 8.535 (5.165–14.106) | <0.001 | 2.977 (1.611–5.499) | <0.001 |
| Quartile 1 (lowest) | 18.548 (10.663–32.266) | <0.001 | 4.099 (2.088–8.048) | <0.001 |
| Fast speed (m/s) † | | | | |
| Quartile 4 (highest) | | | | |
| Quartile 3 | 4.10 (2.482–6.768) | <0.001 | 2.33 (1.266–4.312) | <0.001 |
| Quartile 2 | 8.884 (5.369–14.695) | <0.001 | 3.05 (1.636–5.705) | <0.001 |
| Quartile 1 (lowest) | 13.200 (7.790–22.352) | <0.001 | 3.15 (1.640–6.060) | <0.001 |

OR = odds ratio; [c] = continuous variable; m = meter; s = second; gait usual speed = Q1 < 1.02, Q2 1.02–1.23, Q3 1.24–1.64, Q4 ≥ 1.65; gait fast speed = Q1 < 1.35, Q2 1.35–1.64, Q3 1.65–2.25, Q4 ≥ 2.26; * logistic regression; † multinomial regression, model adjusted for sex, age, MMSE and years of education.

### 3.5. Associations between Usual GS (Quartile) and Cognitive Impairment

Regarding the multinomial analysis (see Table 3 for an overview), the unadjusted model showed significant associations ($p < 0.001$). Older adults classified with usual GS in Q1 and Q2 (lowest) and Q3 (medium) were indicated as 18.55 (CI 10.663–32.266), 8.35 (CI 5.165–14.106) and 3.48 (CI 2.091–5.817) times more likely to develop cognitive impairment, respectively, compared to those ranked in the highest quartile. When the analysis was adjusted for confounders, the associations were attenuated, remaining significant for the lowest quartiles ($p < 0.001$), but not for the medium-velocity quartile ($p > 0.050$). Thus, older adults classified in Q1 and Q2 were indicated as 4.10 (CI 2.088–8.048) and 2.98 (CI 1.611–5.499) times more likely to develop cognitive impairment than those classified in the highest quartile. The analysis showed no significant association for usual GS in Q3 (OR = 1.773, CI 0.953–3.297, $p = 0.071$).

### 3.6. Associations between Fast GS (Quartile) and Cognitive Impairment

Following the results, the unadjusted analysis indicated a higher level of association for fast GS in the lowest-velocity quartiles (see Table 3 for an overview). Older adults classified in Q1, Q2 and Q3 were presented as 13.20 (CI 7.790–22.352), 8.88 (CI 5.369–14.695) and 4.10 (CI 2.482–6.768) times more likely to develop cognitive impairment, respectively, in relation to being ranked in the highest quartile. After controlling for potential confounders (i.e., sex, age, MMSE and years of education), the associations were considerably attenuated, but remained significant ($p < 0.001$). Therefore, participants classified in Q1, Q2 and Q3 were 3.15 (CI 1.640–6.060), 3.05 (CI 1.636–5.705) and 2.33 (CI 1.266–4.312) times more likely to develop cognitive impairment, respectively, when compared to others classified in the highest quartile.

## 4. Discussion

To the best of our knowledge, this is the first population-based study to identify associations between GS and cognitive vulnerability carried out with cognitively normal older Brazilian adults. A strong point of this study is the recruitment of the population, carried out in a defined geographic area (Manaus city). Population-based health information from areas far from large urban centers plays an important role in disease surveillance, management and analysis [39]. Thus, the findings highlight the unknown information about the older Brazilian population of Amazonas, considered comparatively to other states in the country, a region of extreme vulnerability in health and quality of life [40,41].

We confirmed our premise that high GS values would be positively associated with high cognitive performance. Pearson's positive and strong associations were found between the usual and fast measure of GS and cognition, and the results are in agreement with previous empirical research [42–44]. Our second hypothesis was also confirmed: we found that high values of usual and fast GS (as a continuous variable) were able to predict cognitive vulnerability. Elevated GS values indicated a protective role in cognitive performance. From this perspective, through significant associations between usual and fast GS and low cognitive performance, we show that GS can be assumed as a non-invasive biomarker capable of detecting cognitive alterations in cognitively normal older adults [6,7,45].

Previous investigations have highlighted that typical ageing diseases such as Alzheimer's and dementia are progressive [46,47], and the transition from cognitive health from MCI to advanced impairment can take a decade or more [26,48], and even remain undetectable. Thus, the development of useful measures to track cognitive vulnerability in cognitively normal older adults at an early stage is an important and necessary procedure, and could have an important impact on the healthcare field [49]. We found a higher prevalence of low GS in the group with cognitive impairment. This finding is in agreement with previous population studies that found associations between low GS and cognitive vulnerability [9,50–52].

Low GS reflects the possibility that older people have advancing functional and/or structural brain changes [1,5,20]. Based on the results offered by the gait measurement, more in-depth examinations could be performed with participants who indicated low gait values. It is known that a low GS suggests a reduction in grey matter volume in the prefrontal cortex [53,54], changes in the subcortical regions in the basal ganglia or cerebellum [17], a reduction in periventricular and subcortical white matter [53], atrophy of medial temporal areas [30], as well as a reduction in hippocampal volumes [54]. Moreover, our associations corroborate the proposal of a "motor signature" of cognitive decline [48,55] and also with the syndrome of "Cognitive–Motor Risk" [53]. Both theories indicate that, in advanced age, idling is able to confirm complaints of cognitive impairment, which may be associated with pre-dementia syndromes, therefore indicating potential risks for progressing to dementia [7].

By analyzing the usual GS as a continuous variable, we found that each increase in standard deviation (SD) of 0.1 m/s indicated a chance of improvement in cognitive performance of up to 55%. After controlling for potential confounders (i.e., sex, age, MMSE and years of education), for each increase in standard deviation (SD) of 0.1 m/s, the chance of developing a cognitive deficit was attenuated by up to 82%. Regarding fast GS as a continuous variable, for each increase in the standard deviation (SD) of 0.1 m/s, there was a chance of improvement in cognitive performance of up to 57%. After controlling for potential confounders (i.e., sex, age, MMSE and years of education), for each increase in standard deviation (SD) of 0.1 m/s from fast GS, cognitive impairment was attenuated by up to 85%. One interpretation of this result is that having a preserved cognitive status (MMSE assessment) and longer years of education acts as a protective factor, increasing the individual's chance of not developing cognitive decline [9,29,56]. Moreover, it should be considered that GS is not an exclusive predictor of future cognitive impairment at an advanced age. The process has multifaceted correlates, including age, sex, anthropometry, genetics, diet, past events and lifestyle [55,57].

The analysis in quartiles offered a more detailed understanding of the potential of the GS exam to predict the chance of cognitive vulnerability: a second hypothesis. Thus, older adults classified with usual GS in Q1, Q2 and Q3 indicated a chance of presenting cognitive vulnerability of 175%, 75% and 24%, respectively, in relation to those classified in the highest quartile (Q4). By controlling for potential confounders (i.e., sex, age, MMSE and years of education), the levels of association decreased considerably, but even so, participants classified in Q1 (slowest) and Q2 indicated a chance of developing cognitive impairment of up to 30% and 19%, respectively. For Q3 (1.24–1.64 m/s), there was no significant association. A possible explanation for this is that the performance of these participants (Q3) was equal to or superior to the performance of a usual GS (1.2 m/s to

1.4 m/s), considered in the literature as normal/preserved gait [29], while a GS below 0.8 m/s is associated with impaired mobility [57].

These findings are in line with previous evidence [48–50]. Regarding fast GS, our analysis showed, for Q1, Q2 and Q3, an increase in the chance of presenting a cognitive deficit of 122%, 78% and 31%, respectively, in relation to the participants classified in the highest quartile (Q4). Controlling for potential confounders (i.e., sex, age, MMSE and years of education), the chance remained higher in Q1 and Q2, respectively, with rates up to 21% and 20%. For those with speed rated in Q3 (medium), the chance of experiencing cognitive vulnerability was up to 13%, and therefore lower than in Q1 and Q2 (poor speed). These results confirmed the usual and fast GS as potential screening measures (biomarkers) of cognitive vulnerability in cognitively normal older adults, a potentially useful strategy to prevent cognitive impairment [6,8,9], as well as crucial for maintaining autonomy [7].

*Limitations*

Our study had the following limitations. First, the cross-sectional results are not directly generalizable to changes over time and to other populations. Second, we focused the analysis on only one temporal parameter of gait (speed). Therefore, we did not assess other important gait parameters (e.g., pace, stride length, double stance time, cadence, gait stability ratio). Finally, we found some barriers to fully drawing comparisons between our results and previous studies due to their methodological differences, such as heterogeneity between the protocols and instruments used to assess cognition and gait, as well as the disproportionate numbers of men and women and inequality in the years of education for each population.

## 5. Conclusions

Our population-based study showed that slow GS values considerably increase the chance of cognitive vulnerability in older adults. The results were consistent with the hypothesis that slow GS can be considered a predictor of cognitive deficits even in the cognitively normal older population [6,27,28]. The associations found between cognition and GS arise because gait demands the sharing of external information with internal neural networks. This requires the integration of different cognitive domains (e.g., attention, planning, visuospatial and motor processes). Our findings corroborate the thesis that understanding the functionality of gait and cognition together may be superior to understanding their constructs only separately [7], and they suggest that the examination of the GS can help to elucidate the cognitive integration of multiple and complex brain processes. This study has remarkable implications for public health and clinical practice: GS assessment is an attractive screening measure; in a short time and at a low cost (stopwatch and cutoff point), it is possible to perform routine clinical examinations. It is suggested that further investigations be carried out and that these include longitudinal follow-up to qualify the understanding of GS as a biomarker of cognitive decline in cognitively normal older adults.

**Author Contributions:** Conceptualization, M.d.M.N., É.R.G., A.M. and A.I.; methodology, M.d.M.N. and É.R.G.; software, É.R.G.; validation, M.d.M.N., É.R.G., A.M., P.M. and B.R.G.; formal analysis, M.d.M.N.; investigation, M.d.M.N., É.R.G., B.R.G., A.M. and A.I.; resources, É.R.G., B.R.G. and A.I.; data curation, M.d.M.N. and É.R.G.; writing—original draft preparation, M.d.M.N., É.R.G. and A.I.; writing—review and editing, A.M., B.R.G. and P.M.; visualization, M.d.M.N., A.M., P.M. and É.R.G.; supervision, É.R.G., B.R.G. and A.I.; project administration, É.R.G., B.R.G., A.M. and A.I.; funding acquisition, É.R.G., B.R.G., A.M. and A.I. All authors have read and agreed to the published version of the manuscript.

**Funding:** We acknowledge support from the Swiss National Centre of Competence in Research LIVES—Overcoming vulnerability: life course perspectives, which is funded by the Swiss National Science Foundation (grant number: 51NF40-185901). Moreover, A.I. acknowledges support from the Swiss National Science Foundation (grant number: 10001C_189407). É.R.G. and B.R.G. acknowledge support from LARSyS—Portuguese national funding agency for science, research and technology (FCT) pluriannual funding 2020–2023 (Reference: UIDB/50009/2020).

**Institutional Review Board Statement:** The study was conducted according to the guidelines of the Declaration of Helsinki and had been approved by the local ethics committee before the start of the data collection (ethic committee name: The Research Ethics Committee—Human Beings; approval code: CAAE: 56519616.6.0000.5016, Number: 1.599.258, Brazil Platform; approval date: 20 June 2016).

**Informed Consent Statement:** Informed consent was obtained from all subjects involved in the study before participation.

**Data Availability Statement:** The data presented in this study are available upon request from the corresponding author.

**Acknowledgments:** The authors are grateful to Duarte L. Freitas and Jefferson Jurema for their help in setting up the study, as well as Maria A. Tinôco, Floramara T. Machado, Angenay P. Odim and Bárbara R. Muniz for the technical assistance in the data collection and management. We are especially grateful to the older people for their participation and interest.

**Conflicts of Interest:** The authors declare no conflict of interest. The funders had no role in the design of the study; in the collection, analyses or interpretation of data; in the writing of the manuscript or in the decision to publish the results.

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
