# Peer review of "Gait Speed as a Biomarker of Cognitive Vulnerability: A Population-Based Study with Cognitively Normal Older Adults"

_sustainability, doi:10.3390/su14127348_

Round 1

Reviewer 1 Report

The Authors present an interesting paper on the association of Gait Speed and cognitive functioning.

In my opinion, in the Introduction the Gait Speed should be explained more: it's an easy test, performable in the outpatient setting, with multiple uses, as an example, the Authors could cite the Comprehensive Geriatric Assessment used in Oncology to determine patient fitness.

I believe that the 30 and 31 reference are to be switched, considering the COGTEL is first mentioned in line 117.

Given the vast difference in age between the groups (as one should expect, older people tend to have more cognitive impairment), why the age was not controlled as a confounder in the multivariate analysis?

Cancer diagnosis and treatment has a significant impact on both gait speed and cognitive function: why wasn't considered in the "comorbidities"?

Line 332 should read : Q1 and Q2

Author Response

Dear Reviewer:
1. Below are the answers to your considerations (we appreciate your contributions);
2. "Please see the attachment" the manuscript with the requested changes.

  1. In my opinion, in the Introduction the Gait Speed should be explained more: it's an easy test, performable in the outpatient setting, with multiple uses, as an example, the Authors could cite the Comprehensive Geriatric Assessment used in Oncology to determine patient fitness.

Reply:

Dear Reviewer, thank you for your helpful remarks. Following your suggestions, in the Introduction section, information on gait assessment (advantages) has been expanded (page 2, lines 65-69).

  1. I believe that the 30 and 31 reference are to be switched, considering the COGTEL is first mentioned in line 117.

Reply:

Dear Reviewer, thank you. The Cogtel test references have been adjusted (page 3, lines 125-137).

  1. Given the vast difference in age between the groups (as one should expect, older people tend to have more cognitive impairment), why the age was not controlled as a confounder in the multivariate analysis?

Reply:

Dear Reviewer, thank you for your helpful remarks. We inform that the variables "sex and age" were introduced in the multivariate analysis. As suggested, we now include the variables "sex and age" together with the variables "MMSE and years of education" throughout the text in Table 3, as well as in the presentation of these Results, and finally in the Discussion section. All 4 covariates had already been mentioned in the section "2.2.3. Covariates" (page 3, line 142).

  1. Cancer diagnosis and treatment has a significant impact on both gait speed and cognitive function: why wasn't considered in the "comorbidities"?

Reply:

Dear Reviewer, information on "cancer diagnosis and treatment" was present in the SEVAAI project comorbidity questionnaire. However, among the 701 participants, none reported the presence of cancer or having undergone treatment for this disease.

  1. Line 332 should read : Q1 and Q2.

Reply:

Dear Reviewer, thank you, the typo has been corrected.

Reviewer 2 Report

I think it is abundantly clear to all those in this field that gait and cognition  are fundamental features of a complex and syndromic scenario of aging and frailty.  Although this work is commendable, it is honestly unclear to me what real, potential, impact this would have on the field. Several investigations, structured and omni-comprehensive, are the current and foreseeable trends, and therefore I am not sure if there really is a necessity for this work. In other words, this study-design relies on a sole association that is not only well ascertained and expected, but it runs the risk to result as a simplistic one. The Authors themselves have acknowledged such extent in their limitations.

In conclusion, issues with the significance of the work and reporting are too great to overcome. Certainly this point warrants addressing and, in the current version, this reviewer recommends major revisions.

Author Response

Dear Reviewer:
1. Below are the answers to your considerations (we appreciate your contributions);
2. "Please see the attachment" the manuscript with the changes requested by the other Reviewers.

Reply

Dear Reviewer, thank you for your helpful remarks. We now better clarify the different gaps that our study will address. Specifically, few studies have examined the association between GS and cognitive vul-nerability in cognitively normal older adults, especially in a population-based study [6]. The most cross-sectional investigations have included older adults with cognitive impairments (e.g., Parkinson's disease, Alzheimer's, dementia). Moreover, due to methodological differences between existing cross-sectional studies, there is a low consensus on the associations between cognitive impairment and GS, especially in the single-task condition [22] [26]. When dealing with the older and cognitively normal population, cross-sectional studies have the advantage of helping to identify cognitive alterations during the genesis of the process (pre-clinical phases), a propitious moment for an early therapeutic intervention. In addition, there is a lack of information on the relationship between GS and cognitive change in normal aging, especially in older Latin American populations [8].

Although, in recent decades, studies have examined the association between GS and cognitive impairment with cognitively normal older adults from different coun-tries, such as the United States (Fitzpatrick et al., 2007; Mielke et al., 2013; Savica et al., 2016), Canada (Montero-Odasso et al., 2018; Kueper et al., 2020), and Japan (Taniguchi et al., 2011), to date, we are not aware of population-based investigations that have investigated associations between GS and cognitive deficits in cognitively normal and independent Brazilian older adults. Information based on the geolocation of the cohort is important, as it provides a deeper understanding of the facets of the ageing process, contributing to the creation or restructuring of specific health policies/strategies. Ac-cording to the Brazilian Society of Geriatrics and Gerontology [23], compared to other regions of the country, the older population of the North region is ageing in conditions of greater vulnerability: the process takes place in a scenario of socio-economic fragili-ty, with worse access to basic care and education.

Thus, we used data from the Health, Lifestyle and Physical Fitness in Adults and Elderly in Amazonas (SEVAAI) research project to examine associations between cog-nitive vulnerability and GS in a large older sample. We hypothesize that (1) higher GS values are positively associated with better cognitive performance, and (2) low GS is associated to cognitive vulnerability in cognitively normal older adults.

Reviewer 3 Report

I found the article very interesting to read. Its conclusions are very interesting for early impairment assessment, prevention and early intervention.

Here are some constructive comments to improve the publication.

Introduction.

The information and data are presented in a clear and concise manner. In the text it says: "To date, we are not aware of population-based investigations that have investigated associations between GS and cognitive deficits in cognitively normal and independent Brazilian older adults. ". The only comment I could make is whether the authors are aware of other research on the relationship between gait speed and cognitive vulnerability in cognitively normal and independent older adults, outside Brazil.

RESULTS.

Table 1 shows that the group of people with cognitive impairment has an average of 0 years of education, although the IQR is 0-16. Is this data correct? If so, it would be worth explaining in the text because the difference between this group and the one showing normal cognitive functioning in the variable "years of education" could be somehow affecting the differences on cognitive performance.

Author Response

Dear Reviewer:
1. Below are the answers to your considerations (we appreciate your contributions);
2. "Please see the attachment" the manuscript with the requested changes.

  1. The information and data are presented in a clear and concise manner. In the text it says: "To date, we are not aware of population-based investigations that have investigated associations between GS and cognitive deficits in cognitively normal and independent Brazilian older adults". The only comment I could make is whether the authors are aware of other research on the relationship between gait speed and cognitive vulnerability in cognitively normal and independent older adults, outside Brazil.

Reply:

Dear Reviewer, thank you for this helpful remark. Information on previous studies carried out in other countries with the same focus as our investigation has been included in the introduction section (page 2, lines 87-90).

  1. RESULTS.

Table 1 shows that the group of people with cognitive impairment has an average of 0 years of education, although the IQR is 0-16. Is this data correct? If so, it would be worth explaining in the text because the difference between this group and the one showing normal cognitive functioning in the variable "years of education" could be somehow affecting the differences on cognitive performance.

Reply

Dear Reviewer,

a) To clarify, we have reviewed our database and the information in Table 1 about the "years of education" of both groups, and these are correct;

b) Regarding an explanation in the text about "years of education" and the effects of this variable on the results, we took care to control for this education variable during the analyses (page 3, lines 144);

c) Regarding a better explanation of the fact, in the Discussion section (page 9, lines 324-326), we present a causal implication of "years of education" for cognitive performance.

Round 2

Reviewer 1 Report

I am satisfied with the changes in the manuscript.

Reviewer 2 Report

Although I reckon the Authors' replies only partially addressing the issues raised, the works remains peculiar of the population investigated and it is worth to be communicated.